# *Storyboarding* OF RECIPES: GROUNDED CONTEXTUAL GENERATION

**Anonymous, Anonymous & Anonymous**
Anonymous
Anonymous
`{anonymous, anonymous, anonymous,}@anonymous`

## ABSTRACT

Information need of humans is essentially multimodal in nature, enabling maximum exploitation of situated context. We introduce a dataset for sequential procedural (*how-to*) text generation from images in cooking domain. The dataset consists of 16,441 cooking recipes with 160,479 photos associated with different steps. We setup a baseline motivated by the best performing model in terms of human evaluation for the Visual Story Telling (ViST) task. In addition, we introduce two models to incorporate high level structure learnt by a Finite State Machine (FSM) in neural sequential generation process by: (1) Scaffolding Structure in Decoder (SSiD) (2) Scaffolding Structure in Loss (SSiL). These models show an improvement in empirical as well as human evaluation. Our best performing model (SSiL) achieves a METEOR score of 0.31, which is an improvement of 0.6 over the baseline model. We also conducted human evaluation of the generated grounded recipes, which reveal that 61% found that our proposed (SSiL) model is better than the baseline model in terms of overall recipes, and 72.5% preferred our model in terms of coherence and structure. We also discuss analysis of the output highlighting key important NLP issues for prospective directions.

## 1 INTRODUCTION

Interpretation is heavily conditioned on context. Real world interactions provide this context in multiple modalities. In this paper, the context is derived from vision and language. The description of a picture changes drastically when seen in a sequential narrative context. Formally, this task is defined as: given a sequence of images $\mathbb{I} = \{I_1, I_2, ..., I_n\}$ and pairwise associated textual descriptions, $\mathbb{T} = \{T_1, T_2, ..., T_n\}$; for a new sequence $\mathbb{I}'$, our task is to generate the corresponding $\mathbb{T}'$. Figure 1 depicts an example for making *vegetable lasagna*, where the input is the first row and the output is the second row. We call this a *'storyboard'*, since it unravels the most important steps of a procedure associated with corresponding natural language text. The sequential context differentiates this task from image captioning in isolation. The narration of procedural content draws slight differentiation of this task from visual story telling. The dataset is similar to that presented by ViST Huang et al. (2016) with an apparent difference between stories and instructional in-domain text which is the clear transition in phases of the narrative. This task supplements the task of ViST with richer context of goal oriented procedure (*how-to*). This paper attempts at capturing this high level structure present in procedural text and imposing this structure while generating sequential text from corresponding sequences of images.

Numerous online blogs and videos depict various categories of *how-to* guides for games, do-it-yourself (DIY) crafts, technology, gardening etc. This task lays initial foundations for full fledged storyboarding of a given video, by selecting the right junctions/clips to ground significant events and generate sequential textual descriptions. However, the main focus of this work is generating text from a given set of images. We are going to focus on the domain of cooking recipes in the rest of this paper, leaving the exploration of other domains to future. The two important dimensions to address in text generation are content and structure. In this paper, we discuss our approach in generating more structural/coherent cooking recipes by explicitly modeling the state transitions between different stages of cooking (phases). We address the question of generating textual interpretation of

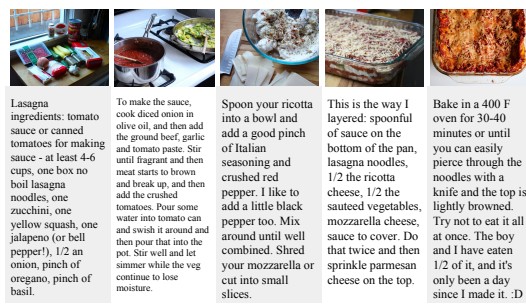

Figure 1: Storyboard for the recipe of vegetable lasagna

the procedure depicted as a sequence of pictures (snapped at different instances of time as the procedure progresses). We introduce a framework to apply traditional FSMs to enhance incorporation of structure in neural text generation. We plan to explore backpropable variants in place of FSMs in future to design structure aware generation models.

The two main contributions of this paper are:

1. A dataset of 16k recipes targeted for sequential multimodal procedural text generation.
2. Two models (SSiD: Structural Scaffolding in Decoder ,and SSiL: Structural Scaffolding in Loss) for incorporating high level structure learnt by an FSM into a neural text generation model to improve structure/coherence.

The rest of the paper is organized as follows. Section 2 describes the related work performed along the lines of planning while generating, understanding food and visual story telling. Section 3 describes the data we gathered for this task and related statistics. In Section 4, we describe our models: a baseline model (Glocal), SSiD and SSiL in detail. Section 5 presents the results attained by each of these models both empirically and qualitatively. Section 6 concludes this work and presents some future directions.

## 2 RELATED WORK

**Why domain constraint?** Martin et al. (2017) and Khalifa et al. (2017) demonstrated that the predictive ability of a seq2seq model improves as the language corpus is reduced to a specialized domain with specific actions. Our choice of restricting domain to recipes is inspired from this, where the set of events are specialized (such as 'cut', 'mix', 'add') although we are not using event representations explicitly. These specialized set of events are correlated to phases of procedural text as described in the following sections.

**Planning while writing content:** A major challenge faced by neural text generation Lu et al. (2018) while generating long sequences is the inability to maintain structure, contravening the coherence of the overall generated text. This aspect was also observed in various tasks like summarization Liu et al. (2018), story generation Fan et al. (2019). Pre-selecting content and planning to generate accordingly was explored by Puduppully et al. (2018) and Lukin et al. (2015) in contrast to generate as you proceed paradigm. Fan et al. (2018) adapt a hierarchical approach to generate a premise and then stories to improve coherence and fluency. Yao et al. (2018) experimented with static and dynamic schema to realize the entire storyline before generating. However, in this work we propose a hierarchical multi task approach to perform structure aware generation.

**Comprehending Food:** Recent times have seen large scale datasets in food, such as Recipe1M Marin et al. (2018), Food-101 Bossard et al. (2014) and bench-marking challenges like iFood challenge [1]. Food recognition Arora et al. (2019) addresses understanding food from a vision perspective. Salvador et al. (2018) worked on generating cooking instructions by inferring ingredients from an image. Zhou et al. (2018) proposed a method to generate procedure segments for YouCook2 data. In NLP domain, this is studied as generating procedural text by including ingredients as checklists

---

[1] https://www.kaggle.com/c/ifood2018

Kiddon et al. (2016) or treating the recipe as a flow graph Mori et al. (2014). Our work is at the intersection of two modalities (language and vision) by generating procedural text for recipes from a sequence of images. Bosselut et al. (2017) worked on reasoning non-mentioned causal effects thereby improving the understanding and generation of procedural text for cooking recipes. This is done by dynamically tracking entities by modeling actions using state transformers.

**Visual Story Telling:** Research at the intersection of language and vision is accelerating with tasks like image captioning Hossain et al. (2019), visual question answering Wu et al. (2017), visual dialog Das et al. (2017) , Mostafazadeh et al. (2017), de Vries et al. (2017), de Vries et al. (2018)). *ViST* Huang et al. (2016) is a sequential vision to language task demonstrating differences between descriptions in isolation and stories in sequences. Along similar lines, Gella et al. (2018) created VideoStory dataset with videos posted on social media with the task of generating a multi-sentence story captions for them. Smilevski et al. (2018) proposed a late fusion based model for ViST challenge. Kim et al. (2018) attained the highest scores on human readability in this task by attending to both global and local contexts. We use this as our baseline model and propose two techniques on top of this baseline to impose structure needed for procedural text.

## 3 DATASET DESCRIPTION

Food is one of the most photographed subject on the instagram network which led to coining the term *foodstagram*. We identified two *how-to* blogs: *instructables*[2] and *snapguide.com*[3], comprising step-wise instructions (images and text) of various *how-to* activities like games, crafts etc,. We gathered 16,441 samples with 160,479 photos [4] for food, dessert and recipe topics. We used 80% for training, 10% for validation and 10% for testing our models. In some cases, there are multiple images for the same step and we randomly select an image from the set of images. We indicate that there is a potential space for research here, in selecting most distinguishing/representative/meaningful image. Details of the datasets are presented in Table 1. The distribution of the topics is visualized here[5]. A trivial extension could be done on other domains like gardening, origani crafts, fixing guitar strings etc, which is left for future work.

| Data Sources | # Recipes | # Avg Steps |
|---|---|---|
| *instructables* | 9,101 | 7.14 |
| *snapguide* | 7,340 | 13.01 |

Table 1: Details of dataset for *storyboarding* recipes

## 4 MODEL DESCRIPTION

We first describe a baseline model for the task of storyboarding cooking recipes in this section. We then propose two models with incremental improvements to incorporate the structure of procedural text in the generated recipes : SSiD (Scaffolding Structure in Decoder) and SSiL (Scaffolding Structure in Loss). The architecture of *scaffolding* structure is presented in Figure 2, of which different aspects are described in the following subsections.

### 4.1 BASELINE MODEL (GLOCAL):

We have a sequence of images at different phases of cooking as our input and the task is to generate step wise textual descriptions of the recipe.

The baseline model is inspired from the best performing system in ViST challenge with respect to human evaluation Kim et al. (2018). The images are first resized into 224 X 224. Image features for each step are extracted from the penultimate layer of pre-trained ResNet-152 He et al. (2016). These features are then passed through an affinity layer to obtain an image feature of dimension 1024.

---

[2] https://www.instructables.com/

[3] https://snapguide.com/

[4] ** *data will be released* **

[5] https://storyboarding.github.io/story-boarding/

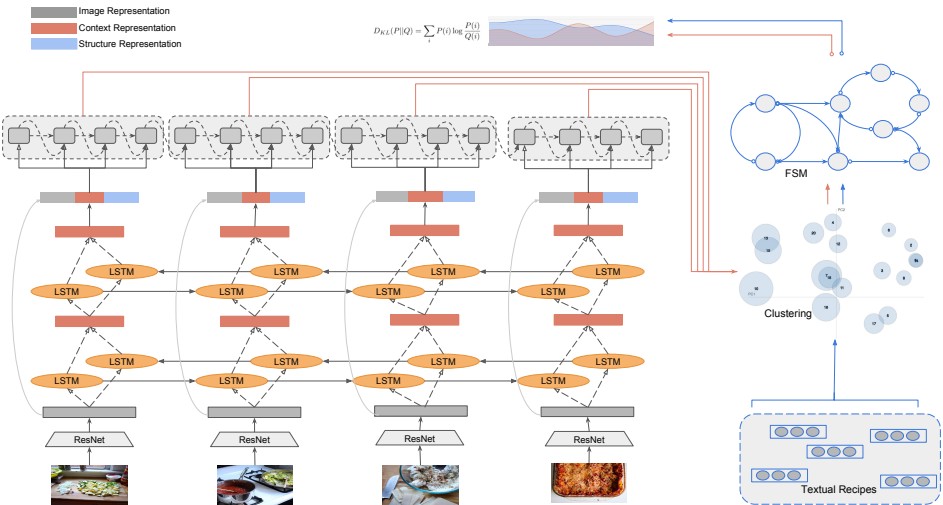

Figure 2: Architecture for incorporating high level structure in neural recipe generation

To maintain the context of the entire recipe (global context), the sequence of these image features are passed through a two layered Bi-LSTM with a hidden size of 1024. To maintain specificity of the current image (local context), the image features for the current step are concatenated using a skip connection to the output of the Bi-LSTM to obtain glocal representation. Dropout of 0.5 is applied systematically at the affinity layer to obtain the image feature representation and after the Bi-LSTM layer. Batch normalization is applied with a momentum 0.01. This completes the encoder part of the sequence to sequence architecture. These glocal vectors are used for decoding each step. These features are passed through a fully connected layer to obtain a representation of 1024 dimension followed by a non-linear transformation using ReLU. These features are then passed through a decoder LSTM for each step in the recipe which are trained by teacher forcing. The overall coherence in generation is addressed by feeding the decoder state of the previous step to the next one. This is a seq2seq model translating one modality into another. The model is optimized using Adam with a learning rate of 0.001 and weight decay of 1e-5.

The model described above does not explicitly cater to the structure of the narration of recipes in the generation process. However, we know that procedural text has a high level structure that carries a skeleton of the narrative. In the subsequent subsections, we present two models that impose this high level narrative structure as a scaffold. While this scaffold lies external to the baseline model, it functions on imposing the structure in decoder (SSiD) and in the loss term (SSiL).

## 4.2   SCAFFOLDING STRUCTURE IN DECODER (SSiD):

There is a high level latent structure involved in a cooking recipe that adheres to transitions between steps, that we define as *phases*. Note that the steps and phases are different here. To be specific, according to our definition, one or more steps map to a phase (this work does not deal with multiple phases being a part of a single step). Phases may be 'listing ingredients', 'baking', 'garnishing' etc., The key idea of the SSiD model is to incorporate the sequence of phases in the decoder to impose structure during text generation [6].

There are two sources of supervision to drive the model: (1) multimodal dataset $\mathbb{M} = \{\mathbb{I}, \mathbb{T}\}$ from Section 3, (2) unimodal textual recipes[7] $\mathbb{U}$ to learn phase sequences. Finer phases are learnt using clustering followed by an FSM.

---

[6]To validate the hypothesis of operating FSM with phases over the neural baseline model we have in place, we first performed proof of concept experiments with the step-wise titles present in our *instructables* dataset. Here, the content words after removal of the stop words for words with high tf-idf values are defined as phases. However, for the actual model, these phases are latent states learnt through an FSM.

[7]www.ffts.com/recipes.htm

**Clustering:** K-Means clustering is performed on the sentence embeddings with compositional n-gram features Pagliardini et al. (2018) on each step of the recipe in $\mathbb{U}$. Aligning with our intuition, when k is 3, it is observed that these clusters roughly indicate categories of *desserts*, *drinks* and *main course foods* (*pizza*, *quesadilla* etc,). However, we need to find out finer categories of the phases corresponding to the phases in the recipes. We use k-means clustering to obtain the categories of these phases. We experimented with different number of phases $P$ as shown in Table 2. For example, let an example recipe comprise of 4 steps i.e, a sequence of 4 images. At this point, each recipe can be represented as a hard sequence of phases $\mathbf{r} = \langle\, p_1, p_2, p_3, p_4\, \rangle$.

**FSM:** The phases learnt through clustering are not ground truth phases. We explore the usage of an FSM to individually model hard and a softer representation of the phase sequences by leveraging the states in an FSM. We first describe how the hard representation is modeled. The algorithm was originally developed for building language models for limited token sets in grapheme to phoneme prediction. The iterative algorithm starts with an ergodic state for all phase types and uses entropy to find the best state split that would maximize the prediction. This is presented in Algorithm 1. As opposed to phase sequences, each recipe is now represented as a state sequence (decoded from FSM) i.e, $\mathbf{r} = \langle s_1, s_2, s_3, s_4 \rangle$ (hard states). This is a hard representation of the sequence of states.

We next describe how a soft representation of these states is modeled. Since the phases are learnt in an unsupervised fashion and the ground truth of the phases is not available, we explored a softer representation of the states. We hypothesize that a soft representation of the states might smooth the irregularities of phases learnt. From the output of the FSM, we obtain the state transition probabilities from each state to every other state. Each state $s_i$ can be represented as $\langle q_{ij} \,\forall\, j \in S \rangle$ (soft states), where $q_{ij}$ is the state transition probability from $s_i$ to $s_j$ and $S$ is the total number of states. This is the soft representation of state sequences.

The structure in the recipe is learnt as a sequence of phases and/or states (hard or soft). This is the structural *scaffold* that we would like to incorporate in the baseline model. In SSiD model, for each step in the recipe, we identify which phase it is in using the clustering model and use the phase sequence to decode state transitions from the FSM. The state sequences are concatenated to the decoder in the hard version and the state transition probabilities are concatenated in the decoder in the soft version at every time step.

At this point, we have 2 dimensions, one is the complexity of the phases ($P$) and the other is the complexity of the states in FSM ($S$). Comprehensive results of searching this space is presented in Table 2. We plan to explore the usage of hidden markov model in place of FSM in future.

---

**Algorithm 1** State Splitting in Weighted FSM (WFSM)

---

**Result:** FSM along with probabilistic state transitions
initialization **while** *Until end criteria* **do**
    Apply FSM to dataset;
    Record which phases in each recipe go through which states in the WFSM;
    For each state find entropy and reverse sort;
    **for** *Each state in order* **do**
        **if** *if found an entropy reducing split* **then**
            **for** *Each each type coming into the state* **do**
                calculate the entropy at each state if the existing one is split;
                  The score is the each entropy times the number of examples going through that state ;
            **end**
        **else**
    **end**
    Once the best split is found, split moving all incoming arcs of that type to the new state (subtracting them from old one).
**end**

---

## 4.3 SCAFFOLDING STRUCTURE IN LOSS (SSIL):

In addition to imposing structure via SSiD, we explored measuring the deviation of the structure learnt through phase/state sequences from the original structure. This leads to our next model where the deviation of the structure in the generated output from that of the original structure is reflected in the loss. The decoded steps are passed through the clustering model to get phase sequences and then state transition probabilities are decoded from FSM for the generated output. We go a step further to investigate the divergence between the phases of generated and original steps. This can also be viewed as hierarchical multi-task learning Sanh et al. (2018). The first task is to decode each step in the recipe (which uses a cross entropy criterion, $\mathbf{L}_1$). The second task uses KL divergence between

phase sequences of decoded and original steps to penalize the model (say, $\mathbf{L}_2$).When there are $\tau$ steps in a recipe, we obtain $o(s_1^\tau)$ and $g(s_1^\tau)$ as the distributions of phases comprising of soft states for the original and generated recipes respectively. We measure the KL divergence($D_{KL}$) between these distributions:

$$D_{KL}(o(s_1^\tau)||g(s_1^\tau)) = \sum_{i=1}^{\tau} \sum_{j=1}^{S} o(s_i[j]) log \frac{o(s_i[j])}{g(s_i[j])}$$

Each task optimizes different functions and we minimize the combination of the two losses.

$$\sum_{I,T \in \mathbb{I},\mathbb{T}} \mathbf{L}_1(I,T) + \alpha \sum_{U \in \mathbb{U}} \mathbf{L}_2(U)$$

This combined loss is used to penalize the model. Here, $\alpha$ is obtained from KL annealing Bowman et al. (2015) function that gradually increases the weight of KL term from 0 to 1 during train time.

## 5 RESULTS AND DISCUSSION

The two dimensions explored in clustering and FSM are the number of phases that are learnt in unsupervised manner ($P$) and the number of states attained through state splitting algorithm in FSM ($S$). The results of searching this space for the best configuration are presented in Table 2.

| FST Complexity | 1 | 20 | 40 | 60 | 80 | 100 | 120 |
|---|---|---|---|---|---|---|---|
| 20 Phases | 11.27 | 11.60 | 12.31 | 13.71 | 12.32 | 12.51 | 12.36 |
| 40 Phases | 12.03 | 12.44 | 11.48 | 12.58 | 12.50 | **13.91** | 11.82 |
| 60 Phases | 11.13 | 11.18 | 12.74 | 12.26 | 12.47 | 12.98 | 11.47 |

Table 2: BLEU Scores for different number of phases ($P$) and states($S$)

The BLEU score Papineni et al. (2002) is the highest when $P$ is 40 and $S$ is 100. Fixing these values, we compare the models proposed in Table 3. The models with hard phases and hard states are not as stable as the one with soft phases since backpropagation affects the impact of the scaffolded phases. Upon manual inspection, a key observation is that for SSiD model, most of the recipes followed a similar structure. It seemed to be conditioned on a global structure learnt from all recipes rather than the current input. However, SSiL model seems to generate recipes that are conditioned on the structure of each particular example.

| Models | BLEU | METEOR | ROUGE-L |
|---|---|---|---|
| Glocal | 10.74 | 0.25 | 0.31 |
| SSiD (hard phases) | 11.49 | 0.24 | 0.31 |
| SSiD (hard states) | 11.93 | 0.25 | 0.31 |
| SSiD (soft states) | 13.91 | 0.29 | 0.32 |
| **SSiL (soft states)** | **16.38** | **0.31** | **0.34** |

Table 3: Evaluation of storyboarding recipes

**Human Evaluation:** We have also performed human evaluation by conducting user preference study to compare the baseline with our best performing SSiL model. We randomly sampled generated outputs of 20 recipes and asked 10 users to answer two preference questions: (1) preference for overall recipe based on images, (2) preference for structurally coherent recipe. For the second question, we gave examples of what structure and phases mean in a recipe. Our SSiL model was preferred 61% and 72.5% for overall and structural preferences respectively. This shows that while there is a viable space to build models that improve structure, generating an edible recipe needs to be explored to improve the overall preference.

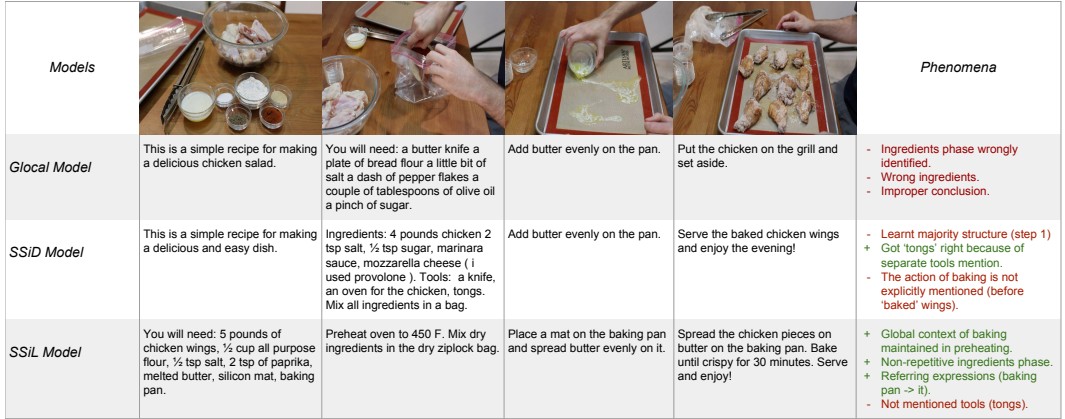

Figure 3: Comparison of generated *storyboards* for *Easy Oven Baked Crispy Chicken Wings*

## 5.1 QUALITATIVE ANALYSIS:

Figure 3 presents the generated text from the three models with an analysis described below.

**Coherence of Referring Expressions:** Introducing referring expressions is a key aspect of coherence Dale (2006), as seen in the case of *'baking pan'* being referred as *'it'* in the SSiL model.

**Context Maintenance:** Maintaining the overall context explicitly has an effect in the generation of each step. This is reflected in SSiL model where *'preheating'* is discussed in second step although the image does not show an oven. This structure is learnt from *baking* step that appears later.

**Schema for Procedural Text:** Explicitly modeling structure for procedural text has enabled the model to conclude the recipe in SSiD and SSiL models by generating words like *'serve'* and *'enjoy'*. Lacking this structure, Glocal model talks about *setting aside* at the end of the recipe.

**Precision of Entities and Actions:** SSiD model introduces *'sugar'* in ingredients after generating *'salt'*. A brief manual examination revealed that this co-occurrence is a common phenomenon. Similarly *sauce* and *cheese* are wrongly generated. SSiL model misses *'tongs'* in the first step.

**Parallels to Summarization:** There is an inherent trade-off between detailing and presenting a concise overview. For instance, one might not need detailing on how onions are cut in comparison to how layering of cheese is executed. Although, we are not explicitly addressing the issue of identifying complicated and trivial steps, a storyboard format implicitly takes care of this by briefing in pictorial representation and detailing in text. This draws parallels with multimodal summarization.

## 6 CONCLUSIONS

Our main focus in this paper is instilling structure learnt from FSMs in neural models for sequential procedural text generation with multimodal data. Recipes are being presented in the form of graphic novels reflecting the cultural change in expectations of presenting instructions. With this change, a storyboard is a comprehensive representation of the important events. In this direction, we gather a dataset of 16k recipes where each step has text and associated images. The main difference between the dataset of ViST and our dataset is that our dataset is targeted at procedural *how-to* kind of text (specifically presenting cooking recipes in this work). We setup a baseline inspired from the best performing model in ViST in the category of human evaluation. We learn a high level structure of the recipe as a sequence of phases and a sequence of hard and soft representations of states learnt from a finite state machine. We propose two techniques for incorporating structure learnt from this as a scaffold. The first model imposes structure on the decoder (SSiD) and the second model imposes structure on the loss function (SSiL) by modeling it as a hierarchical multi-task learning problem. We show that our proposed approach (SSiL) improves upon the baseline and achieves a METEOR score of 0.31, which is an improvement of 0.6 over the baseline model.

We plan on exploring backpropable variants as a scaffold for structure in future. We also plan to extend these models to other domains present in these sources of data. There is no standard way to explicitly evaluate the high level strcuture learnt in this task and we would like to explore evaluation strategies for the same.

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
