# OpenReview forum: "Storyboarding of Recipes: Grounded Contextual Generation"
_ICLR.cc/2019/Workshop/DeepGenStruct — DeepGenStruct 2019_

### Official Review · AnonReviewer1 · 2019-04-11
**Useful new dataset and interesting model, but model not described in enough detail**

**Rating:** 2
**Confidence:** 2

**Review:**

This paper provides:
(1) A new dataset of step-by-step recipes, where each recipe has both an image and a corresponding how-to text. The proposed task is to map from the images to the text.
(2) A description of a model that adds structure to the visual story telling baseline (global-local attention cascading network). They add structure in two ways: (a) SSiD: They cluster sentence representations of recipe sentences into clusters that are called "phases" - thus a recipe can be represented as a sequence of "phases".  Then they learn a weighted FSA which gives the probabilities of transitioning from one phase to another (I think). This is somehow incorporated into the decoder, but I'm not sure how. (b) SSiL: They introduce a loss which says that the structure (i.e. phase sequence) of the images and the generated text must be the same. They show that (a) and (b) improve over the baseline on BLEU, METEOR, ROUGE-L, and human evaluation.

Pros:
- The dataset seems really useful for researchers working on structured multimodal image-to-text generation.
- The overall idea of the SSiD and SSiL models seems like a good idea, and it shows convincing performance improvements.
- There are some really nice visualizations of the phases provided on the dataset website: https://storyboarding.github.io/story-boarding/40topics.html

Cons:
- The SSiD model is not described in nearly enough detail for the reader to understand it properly. See more detailed notes below.
- Reproducibility: Unless I missed something, the authors do not mention whether they will release the code (and the generated output) for their SSiD and SSiL models. Especially given that the models are not explained in enough detail, this severely limits the reproducibility and usefulness of the work. I doubt that any researcher would be able to reproduce the models given only the information in the paper.

Comments about detail of SSiD model:
- There are no equations in section 4.2, thus there is no precise description of the many steps required to build and train the model.
- Though I could mostly understand the first two paragraphs of section 4.2 (though I would not be able to reproduce the results due to the many missing details), I was not able to understand the overall idea of the FSM. For example, what are the states of the FSM? Are the states the phases?
- The FSM is motivated as "softer" but I don't understand how or why it is softer or why softness is a good thing.
- Could you give a brief explanation of "ergodic" when you use it - this would help reader comprehension.
- "These state transition probabilities are concatenated in the decoder." - can you give more detail what you mean here, with an equation?
- Part of the problem is that the baseline model (Glocal) is only very superficially described. Though the reader can of course read the Glocal paper for the details, it would be more accessible if you gave a more detailed description, as the Glocal model is not necessarily well-known or standard. More importantly, this paper should introduce notation in section 4.1 so that section 4.2 can refer to it - to make precise how exactly SSiD works.
- In Algorithm 1, I don't know what many of the lines mean. e.g. "Apply FSM to dataset", "For each state find entropy and reverse sort". Perhaps this is because I didn't understand earlier parts.

Note: You should reference "Simulating Action Dynamics with Neural Process Networks" https://arxiv.org/abs/1711.05313

In conclusion, this dataset looks very useful, the model sounds interesting and seems high-performing, but the model is so insufficiently described that it will frustrate researchers who wish to build on these results. Once the SSiD model is described properly (and the code released), this will be a great paper.

---

### Official Review · AnonReviewer2 · 2019-04-17
**Interesting task and dataset. Good contribution.**

**Rating:** 4
**Confidence:** 2

**Review:**


This paper studies the problem of generating textual descriptions from sequences of images (i.e., a storyboard), by introducing a new dataset and two generation models. The dataset is in the cooking domain, contains about 16k recipes from the how-to blogs, each containing about 7-13 steps on average. The generated descriptions are evaluated with both automatic metrics (Bleu, Meteor, and Rouge-L), and human judgments.

The contribution of this paper is solid, given that the authors will release the datasets. I think this is a very interesting task and will inspire the development of more structure-aware generation models. My main complaint is that the descriptions of the two proposed models (SSiD and SSIL) is relatively vague.

Other questions/ comments:
In Section 4.2, the sequence of phases/states are denoted with sequences of length 4 (e.g. r = <p1, p2, p3, p4>. This is slightly confusing, since there can be many more phases/states, as shown in Table 2.

In the first paragraph of 4.2, the authors mentioned another source of supervision with unimodal textual recipes. How many unimodal recipes are used? Would it be possible to provide more details about them?

---

### Decision · Program_Chairs · 2019-04-19
**Acceptance Decision**

**Decision:**

Accept

**Comment:**

The paper proposes and interesting task/dataset, but the description of the model could be improved. It would be great to take this account for the camera-ready.